# PIKE-A Modulates Mitochondrial Metabolism through Increasing SDHA Expression Mediated by STAT3/FTO Axis

**DOI:** 10.3390/ijms231911304

**Published:** 2022-09-25

**Authors:** Mingming Sun, Qi Yan, Yaya Qiao, Huifang Zhao, Yingzhi Wang, Changliang Shan, Shuai Zhang

**Affiliations:** 1State Key Laboratory of Medicinal Chemical Biology, College of Pharmacy and Tianjin Key Laboratory of Molecular Drug Research, Nankai University, Tianjin 300350, China; 2School of Integrative Medicine, Tianjin University of Traditional Chinese Medicine, Tianjin 301617, China

**Keywords:** PIKE-A, mitochondria, SDHA, cell proliferation, glioblastoma

## Abstract

Previous studies have shown that phosphoinositide 3-kinase enhancer-activating Akt (PIKE-A) is involved in the regulation of several biological processes in cancer. In our previous study, we demonstrated a crucial function of PIKE-A in cancer energy metabolism by regulating pentose phosphate pathway (PPP) flux. However, whether PIKE-A regulates energy metabolism through affecting mitochondrial changes are poorly understood. In the present study, we show that PIKE-A promotes mitochondrial membrane potential, leading to increasing proliferation of glioblastoma cell. Mechanistically, PIKE-A affects the expression of respiratory chain complex Ⅱ succinate dehydrogenase A (SDHA), mediated by regulating the axis of STAT3/FTO. Taken together, these results revealed that inhibition of PIKE-A reduced STAT3/FTO/SDHA expression, leading to the suppression of mitochondrial function. Thus, our findings suggest the PIKE-A/STAT3/FTO/SDHA axis as promising anti-cancer treatment targets.

## 1. Introduction

Glioblastoma is a common and most aggressive primary brain tumor that grows inside the brain parenchyma, often mixed with normal brain tissue [1]. The molecular mechanisms of glioblastoma progression have not yet been clarified. Therefore, finding effective therapeutic targets has become an urgent problem to be solved in the treatment of glioblastoma. Phosphoinositide 3-kinase enhancer-activating Akt (PIKE-A) is a member of the PIKE family, which has three family members (PIKE-S, PIKE-L, and PIKE-A). They are generated by different transcripts and alternative splicing of the gene CENTG1 [2,3,4]. Many reports have described that PIKE-A promotes cancer cell proliferation, invasion, and survival under cellular energy stress conditions, through the association with a multitude binding partners [5,6]. For example, our previous study demonstrated that PIKE-A promotes glioblastoma cancer progression by regulating pentose phosphate pathway (PPP) flux through binding to the signal transducer and activator of transcription 3 (STAT3) [6]. However, whether PIKE-A regulates cancer metabolism through affecting oxidative phosphorylation (OXPHOS) is poorly understood.

Mitochondria are the main sites for cells to perform aerobic respiration and synthesis of adenosine triphosphate (ATP) through electron transfer on the respiratory chain complex [7]. Increasing evidence shows that mitochondria may also be a key player in apoptosis and autophagy, as well as a provider of extranuclear genetic material [8]. Mounting studies show that mitochondrial dysfunction plays a critical role in regulating tumor occurrence and growth [9,10,11,12]. At the molecular, biochemical, metabolic, and genetic levels, the mitochondria of tumor cells are significantly different from those of normal cells. Our previous studies and other group studies have shown that cancer cells exhibit infinite proliferation, resistance to apoptosis, escape from immune surveillance, invasion and metastasis, and abnormal metabolic pathways [13,14,15,16,17,18]. This is also closely related to their mitochondria. The mitochondrial respiratory chain is composed of a series of intermediate transporters that can accept hydrogen or electrons, including NADH dehydrogenase complex, succinate dehydrogenase (SDH), cytochrome bc1 complex, and cytochrome oxidase, which are located in the mitochondria inner membrane. The *SDHA*, *SDHB*, *SDHC*, and *SDHD* genes encode the four subunits of succinate dehydrogenase (SDH), an enzyme involved in the crossroads of the Krebs cycle and the transfer of electrons [19]. It has been demonstrated that SDHA proteins play a pivotal role in cancer cell growth. Myc-mediated SDHA acetylation triggers epigenetic regulation of gene expression and tumorigenesis [20]. However, the role of SDHA in glioblastoma progression have not been well determined.

In the previous study, we have identified a critical role of the PIKE-A/STAT3/G6PD axis in promoting biosynthesis and anti-oxidant defense, which benefits tumor growth and suppression of cellular senescence [10]. In this study, we uncovered novel PIKE-A functions in the promotion of cancer progress through driving mitochondrial dysfunction, which were mediated by regulating the expression of SHDA. Further mechanistic studies showed that PIKE-A was a novel regulator of the STAT3/fat mass- and obesity-associated gene (FTO) axis, which contributed to SDHA expression. Here, we uncover a critical role of the PIKE-A/STAT3/FTO/SDHA axis in promoting mitochondrial metabolism, which was a benefit for tumor growth.

## 2. Results

### 2.1. PIKE-A Promotes Mitochondrial Respiration in Glioblastoma Cells

Mitochondria play a central role in several important cellular processes such as energy production, fatty acid catabolism, and cellular stress response, which support cancer cell growth. To explore the role of PIKE-A in glioblastoma cells from the mitochondria angle, we firstly generated PIKE-A-stable knocked down cells by using lentiviral vectors harboring shRNA (Figure 1A). Then, we assessed the mitochondrial function by analyzing mitochondrial membrane potential. The results showed that PIKE-A knockdown markedly decreased the mitochondrial membrane potential (Figure 1B). Taken together, our data indicate that PIKE-A promotes mitochondrial respiration in glioblastoma cells.

### 2.2. PIKE-A Regulates Mitochondrial Respiration through SDHA

In order to explore the mechanism of PIKE-A in promoting the glioblastoma cells’ mitochondrial respiration, we assessed the metabolic transition before and after knockdown of the PIKE-A by metabolomics. We found that knockdown PIKE-A can significantly promote the accumulation of succinic acid (Figure 2A), which is oxidized by succinate dehydrogenase (SDH) to produce fumaratein in the TCA cycle. SDH is also a subunit of mitochondrial respiratory chain complexes Ⅱ (MRCⅡ) [21,22]. Thus, we explored the activity of MRCⅡ in PIKE-A knockdown cells and found that knockdown PIKE-A decreased the activity of MRCⅡ (Figure 2B).

In humans, there are two catalytic subunits, SDHA and SDHB of SDH, which are located in the mitochondrial matrix, as well as two membrane-spanning subunits, SDHC and SDHD of SDH [22]. In order to investigate the mechanism of PIKE-A and how it regulates MRCⅡ activity through which subunits, we applied RNA-sequence to analyze the differential gene expressions in PIKE-A knockdown and control cells (Figure 2C) and found that PIKE-A knockdown caused the downregulation of SDHA mRNA but did not affect the mRNA levels of SDHB/SDHC/SDHD (Figure 2D). Furthermore, we applied the real-time PCR assay to validate the RNA-sequence data and found that the mRNA levels of SDHA, not SDHB, SDHC, and SDHD were decreased in PIKE-A knockdown cells (Figure 2E). Furthermore, Western blot assay showed that the protein level of SDHA was decreased in PIKE-A knockdown cells (Figure 3C). In summary, our data indicate that PIKE-A regulates mitochondrial respiration through regulating the expression of SDHA.

### 2.3. PIKE-A Increases SDHA Expression through Activating FTO Expression in a STAT3-Dependent Manner

N6-methyladenosine (m6A) is the most abundant methylation modification on mammalian messenger RNA [23]. Fat-mass and obesity-associated protein (FTO), the first RNA demethylase, is involved in the regulation of the major regulatory function in mammals via the regulation of m6A modifications [24]. For example, our previous studies show that FTO can regulate the expression of 6PGD and G6PD by demethylating m6A modification on their mRNA by demethylation [24,25]. Thus, to identify the mechanism of PIKE-A in regulating SDHA expression, we explored whether PIKE-A regulates SDHA expression mediated by FTO. Firstly, we examined the expression of PIKE-A in U251 and LN299 cells (Figure 3A). Then, we detected the expression of FTO in PIKE-A knockdown cells and found that knockdown PIKE-A significantly inhibited FTO expression at the mRNA and protein level (Figure 3B,C). In addition, we overexpressed PIKE-A in U251 cells and discovered that exogenous expression PIKE-A increased the expression of FTO and SDHA in U251 cells (Figure 3D,E). To validate whether PIKE-A regulates SDHA mediated by FTO, we treated the PIKE-A exogenous expression U251 cells with an FTO inhibitor (Rhein), and found that the increased SDHA mediated by PIKE-A was abolished by Rhein (Figure 3F,G). The implication of these results indicated that PIKE-A increased the expression of SDHA by regulating FTO.

Our previous study has demonstrated that phosphorylation levels of STAT3 on the Y705 site was reduced in U251 and LN229 cells with PIKE-A knockdown [6]. To further explore the underlying mechanisms of PIKE-A regulating FTO expression, we determined the effect of STAT3 on the expression of FTO and SDHA and found that exogenous expression of STAT3 increased FTO, SDHA mRNA and protein levels in LN229 cells (Figure 4A–C). Then, we explored whether PIKE-A regulates the expression of FTO and SDHA through STAT3. We treated the exogenous expression of PIKE-A cells with a STAT3 inhibitor (STATTIC), and the result showed that the increased FTO, SDHA mRNA and protein expression mediated by PIKE-A was abolished by STATTIC treatment (Figure 4D–F). In addition, we overexpressed STAT3 in PIKE-A knockdown cells and found that overexpression of STAT3 increased the expression of PIKE-A, FTO and SDHA. We propose that PIKE-A can regulate STAT3 expression, that STAT3 can regulate PIKE-A expression, and that they form a regulatory loop (Figure 4G). Simultaneously, the upregulation of SDHA mRNA and protein expression mediated by STAT3 was also inhibited by FTO inhibitor (Rhein) treatment (Figure 4H–J). These data showed that STAT3 was necessary for PIKE-A to activate FTO; in addition, FTO is essential for regulating SDHA expression.

### 2.4. PIKE-A Regulates Mitochondrial Respiration by Regulating FTO Expression in a STAT3-Dependent Manner

SDH is one of the hubs linking oxidative phosphorylation and the electron transport chain; we also confirmed that PIKE-A affects mitochondrial respiration by regulating SDHA. Next, we asked whether STAT3 and FTO were downstream of PIKE-A in regulating mitochondrial respiration and cell proliferation, and we performed rescue experiments by expressing STAT3 or FTO in PIKE-A knockdown cells. The data showed that in PIKE-A knockdown cells, exogenous expression of STAT3 or FTO rescued the decreased mitochondrial membrane potential, which was decreased by suppressing the expression of PIKE-A (Figure 5A,B). In addition, we explored whether PIKE-A regulates cell proliferation by regulating STAT3/FTO expression, and we found that exogenous expression of STAT3/FTO would rescue the decreased cell proliferation in PIKE-A knockdown U251 cells (Figure 5C,D). Together, these data suggest that PIKE-A plays a role in regulating mitochondrial respiration and promoting cell proliferation, at least in part through STAT3/FTO.

## 3. Discussion

It was reported that PIKE-A is abnormally expressed in a variety of tumors and has been well recognized to play crucial roles in the regulation of tumorigenesis [4,6,26]. Our previous studies have shown that PIKE-A promotes glioblastoma growth and suppresses cellular senescence by reprogramming energy metabolism through regulating PPP flux [6]. In the present study, we identified another crucial role of PIKE-A in reprogramming energy metabolism by regulating mitochondrial membrane potential, thereby promoting tumor proliferation. Here, we found that PIKE-A promoted SDHA expression via increasing FTO expression through activation of STAT3 signaling and maintaining mitochondrial membrane potential, leading to promoting cancer cell proliferation.

Multiple studies reported that mitochondrial dysfunction plays an important role in tumorigenesis [11,12]. Here, we found that pro-oncogene PIKE-A promotes glioblastoma cell proliferation via regulating mitochondrial functions. Tumors have the advantage of metabolic reprogramming during rapid growth to meet the demand of energy. Here, we found that PIKE-A maintains mitochondrial membrane potential for producing ATP, which supports tumor rapid growth. SDH is a critical metabolic enzyme connecting the TCA to the electron transport chain (ETC) as MRCⅡ, which catalyzes the substrate of succinate to furmate. SDHA as a catalytic subunit of SDH, and its expression and activity, are tightly regulated by oncogenes or tumor suppressors. For example, Myc facilitates the acetylation-dependent deactivation of SDHA by SIRT3 deacetylase in an SKP2-mediated degradation manner [20]. SIRT5-mediated desuccinylation of SDHA promotes renal clear cell carcinoma tumorigenesis [27]. In this study, we identified PIKE-A as a novel regulator of SDHA expression.

Furthermore, the molecular mechanism of how PIKE-A regulates SDHA expression to reprogram mitochondrial dysfunction was explored. Our previous studies have showed that STAT3 is a new regulator and binding partner of PIKE-A, and we found that PIKE-A triggered STAT3-mediated gene transcription activity by enhancing its phosphorylation [6]. In the present study, we found that exogenous of STAT3 in PIKE-A knockdown glioblastoma cells could rescue the decreased SDHA expression. Furthermore, upregulated SDHA mRNA and protein expression mediated by PIKE-A were abolished by STATTIC. Next, we found that PIKE-A promotes SDHA expression via activation of STAT3 signaling mediated by FTO. Lastly, the decreased mitochondrial membrane potential and cell proliferation in knockdown PIKE-A cells were rescued by exogenous expression of STAT3/FTO (Figure 5E).

## 4. Materials and Methods

### 4.1. Cell Culture

U251, LN229, and 293T cells were cultured in Dulbecco’s modified Eagle’s medium (Thermo Fisher Scientific, Waltham, MA, USA) with 10% fetal bovine serum (FBS, ExCell Bio, Shanghai, China) at 37 °C and 5% CO_2_. Cells were passaged when they reached 80 to 90% confluence; the cultures were divided 1:3. Experiments were performed when cells were exponentially growing.

### 4.2. Plasmid Construction and Lentivirus Packaging

Exogenous human PIKE-A CDS sequence was cloned into pLVX3 plasmid for Flag-tag at N-terminus. Exogenous human STAT3 CDS sequence was cloned into pCDH plasmid for Flag-tag at N-terminus. Exogenous human FTO CDS sequence was cloned into pCDNA3.1 plasmid for Flag-tag at N-terminus. Plasmids transfected into cells when grown to 80% confluency using PEI Transfection Reagent (Polysciences, Niles, IL, USA) according to the manufacturer′s protocol.

The short interfering RNA (siRNA)-targeting PIKE-A were purchased from GenePharma (GenePharma, Suzhou, China). For siRNA-mediated gene silencing, cells were grown into 60% confluency and transfected with 20 nM siRNA using siRNA-Mate Transfection Reagent according to the manufacturer′s protocol. Two days after transfection, cells were collected for next experimental analysis.

The short hairpin RNA (shRNA) plasmids targeting PIKE-A were purchased from Transheep Biological Corporation (Transheep, Shanghai, China). To establish stable knockdown cells, lentiviral shRNA constructs were co-transfected with viral packaging plasmids (psPAX2 and pMD.2G) into HEK293T cells. Then, 24 h after transfection, cell supernatants were collected on two consecutive days and filtered through a 0.45 μm filter. U251 and LN229 cells were infected with the Lentivirus for 24 h and selected with 2 μg/mL puromycin. The knockdown efficacity was determined by qRT-PCR.siRNA or shRNA sequences for the specific genes are listed in Table 1.

### 4.3. Cell Proliferation Assay

For cell proliferation assays, 1 × 10^4^ cells were seeded in 24-well plates. The cells were harvested at 0, 1, 2, 3, and 4 days after seeding, and the number was recorded to determined cell growth.

### 4.4. Western Blot Analysis

Cells were lysed with NP40 lysis buffer (150 mmol/L NaCl, 10 mmol/L HEPES [pH = 7.0], 1% NP40, 5 mmol/L Na_4_P_2_O_7_, 5 mmol/L NaF, 2 mmol/L Na_3_VO_4_, protease inhibitor) on ice 20 min and then centrifuged at 12,000 rpm for 20 min at 4 °C to collect the supernatant. Protein samples were separated by 8% SDS-PAGE and then transferred to PVDF membranes (Millipore, Burlington, MA, USA). The membranes were blocked with 5% non-fat milk at room temperature for 2 h, and then the primary antibodies were incubated overnight at 4 °C, and the secondary antibodies were incubated for 1 h at room temperature. Signals were detected using chemiluminescent reagents (Millipore, Burlington, MA, USA). The antibodies are listed in Table 2.

### 4.5. Mitochondrial Respiratory Chain Complexes Ⅱ (MRCⅡ) Activity Assay

MRCⅡ activity (Complex II) was determined by monitoring reduced cytochrome c formation using succinate as substrate. In brief, the 0.2 mg sonicated cell lysates were added into assay buffer (10 mmol/L potassium phosphate (pH 7.4), 2 mM EDTA, 0.01% bovine serum albumin (fatty acid free), 0.2 mmol/L ATP, 1 mmol/L KCN, 5 μM rotenone, and 10 mmol/L succinate); then, the changes in 550 nm absorbance (OD550) were monitored at 30 °C using a Microplate Photometer (Thermo Fisher Scientific, MA, USA) after adding 40 μmol/L oxidized cytochrome c.

### 4.6. qRT-PCR 

Cells were harvested and suspended in 1 mL of TRIzol (Invitrogen, CA, USA), and total RNA was purified using standard methods. cDNA was synthesized using the PrimeScript RT reagent Kit (TaKaRa, Dalian, China) according to the manufacturer’s instructions. Real-time quantitative RT-PCR (qRT-PCR) was conducted on a CFX96™ real-time PCR detection system (Bio-Rad, Hercules, CA, USA). Gene expression was calculated using the 2^–ΔΔCt^ method, and actin was used for gene normalization. Primer sequences for the specific and reference genes are listed in Table 3.

### 4.7. Mitochondrial Membrane Potential Detection

The mitochondrial membrane potential was measured by the mitochondrial membrane potential assay kit with JC-1 (Beyotime, Shanghai, China) following the manufacturer’s instructions. When adherent cells were cultured to 80% confluence in 6-well plates; 1 mL of fresh medium was replaced. Then, 1 mL JC-1 working solution was added, and the cells were incubated in a cell incubator for 20 min. The cells were washed with JC-1 staining buffer twice and then resuspended in 2 mL fresh medium. The experiments were performed according to the manufacturer’s instructions The data were analyzed by Immunofluorescence microscope.

### 4.8. RNA-Sequence

LN229 control and PIKE-A knockdown cells were harvested and suspended in 1 mL of TRIzol (Invitrogen, Carlsbad, CA, USA). Then, dry ice was used to transport to Lc-Bio Technologies Co., Ltd. (Hangzhou, China), as described previously [28].

### 4.9. Metabolite Analysis by Gas Chromatography—Mass Spectrometry

We performed targeted metabolomic analysis of the TCA cycle using high-performance liquid chromatography–tandem mass spectrometry (UPLC-MS/MS) at Shanghai Applied Protein Technology Co., Ltd. as described in Wu et al. [29].

### 4.10. Statistics

Statistical details of experiments include statistical tests, mean ± SD. The differences between two groups with similar variances were analyzed using a two-tailed Student’s *t* test. Comparison among more than two groups was determined by one-way analysis of variance (ANOVA) with Tukey’s post hoc test. All statistical analyses were performed using GraphPad Prism 8 software (GraphPad Software Inc., San Diego, CA, USA). The *p* value lower than 0.05 was considered as significant.

## 5. Conclusions

Taken together, the present study reveals a novel mechanism by which PIKE-A promotes tumor growth and demonstrates that PIKE-A regulates SDHA expression by promoting the STAT3/FTO axis, thereby promoting mitochondrial function and cell proliferation in glioblastoma. In conclusion, the PIKE-A/STAT3/FTO/SDHA pathway might play a pivotal role in glioblastoma conformation, progression, and proliferation. Collectively, our findings suggested that the PIKE-A/STAT3/FTO/SDHA axis is a promising target for anti-glioblastoma therapy.

## Figures and Tables

**Figure 1 ijms-23-11304-f001:**
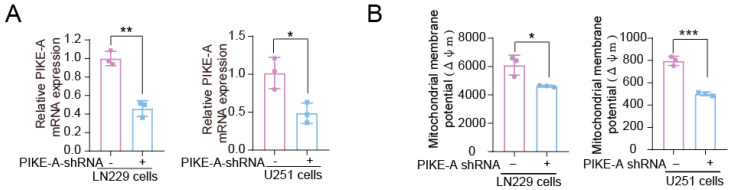
PIKE-A promotes mitochondrial respiration in glioblastoma cells. (**A**) PIKE-A knockdown efficiency in LN229 and U251 cells were determined by real-time PCR (n = 3). The *p* values were calculated using Student’s *t* test. (**B**) PIKE-A knockdown and control cells were tested for mitochondrial membrane potential (n = 3). The *p* values were calculated using Student’s *t* test. Error bars represent mean values ± SD from three independent experiments (*: *p* < 0.05; **: *p* < 0.01; ***: *p* < 0.001).

**Figure 2 ijms-23-11304-f002:**
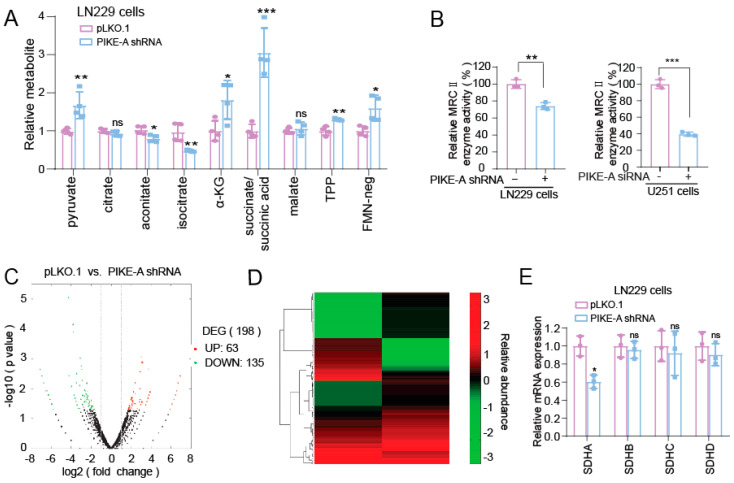
PIKE-A regulates mitochondrial respiration through SDHA. (**A**) Metabolomics was used to detect the changes of various metabolites in control and PIKE-A knockdown cells (n = 4). The *p* values were calculated using Student’s *t* test. (**B**) The enzyme activity of MRCⅡ was measured in PIKE-A knockdown and vector control LN229 and U251 cells (n = 3). The *p* values were calculated using Student’s *t* test. (**C**,**D**) RNA array analysis differential genes in PIKE-A knockdown and vector control LN229 cells. (**E**) SDHA, SDHB, SDHC, and SDHD mRNA levels were analyzed in RNA array between PIKE-A knockdown and vector control LN229 cells (n = 3). The *p* values were calculated using Student’s *t* test. Error bars represent mean values ± SD from three independent experiments (ns: not significant; *: *p* < 0.05; **: *p* < 0.01; ***: *p* < 0.001).

**Figure 3 ijms-23-11304-f003:**
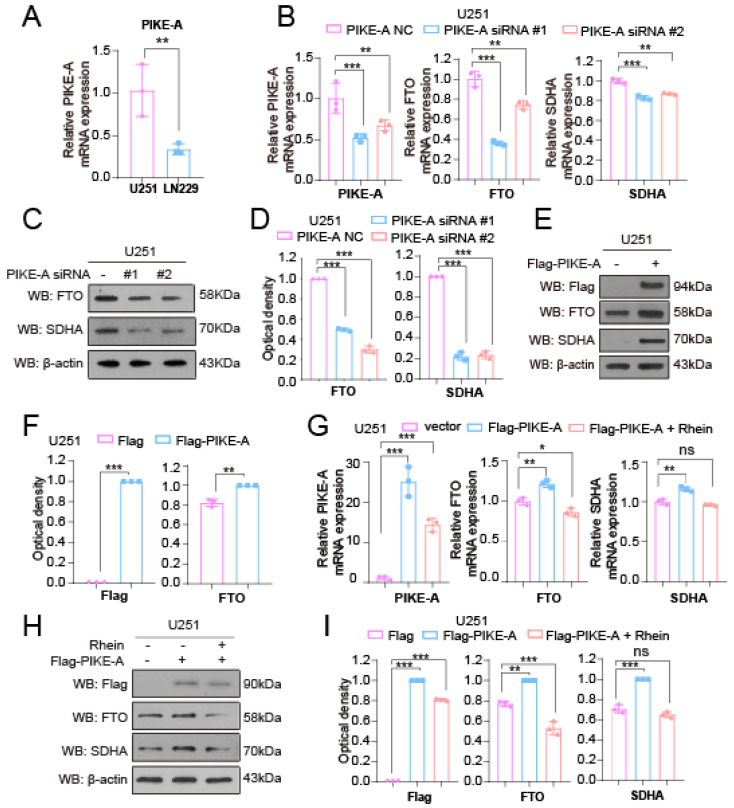
PIKE-A increases SDHA expression through activating FTO expression. (**A**) PIKE-A mRNA levels were determined by qRT-PCR in U251 and LN229 cells (n = 3). The *p* values were calculated using Student’s *t* test. (**B**) FTO and SDHA mRNA levels were determined by qRT-PCR in control and PIKE-A knockdown U251 cells (n = 3). The *p* values were calculated using ANOVA. (**C**,**D**) FTO and SDHA protein levels were determined by Western blotting in control and PIKE-A knockdown U251 cells. (**C**) The optical density of blotting bands was quantified by Image J software and normalized to actin (n = 3). The *p* values were calculated using ANOVA (**D**). (**E**,**F**) FTO and SDHA protein levels were determined by Western blotting in control and PIKE-A overexpressed U251 cells. (**E**) The optical density of blotting bands was quantified by Image J software and normalized to actin (n = 3). The *p* values were calculated using Student’s *t* test (**F**). (**G**) PIKE-A, FTO, and SDHA mRNA levels were determined by qRT-PCR in PIKE-A overexpressed cells treated with Rhein (n = 3). The *p* values were calculated using ANOVA. (**H**,**I**) PIKE-A, FTO, and SDHA protein levels were determined by Western blotting in PIKE-A overexpressed cells treated with Rhein (**H**), The optical density of blotting bands was quantified by Image J software and normalized to actin (n = 3). The *p* values were calculated using ANOVA (**I**). Error bars represent mean values ± SD from three independent experiments ( ns: not significant; *: *p* < 0.05; **: *p* < 0.01; ***: *p* < 0.001).

**Figure 4 ijms-23-11304-f004:**
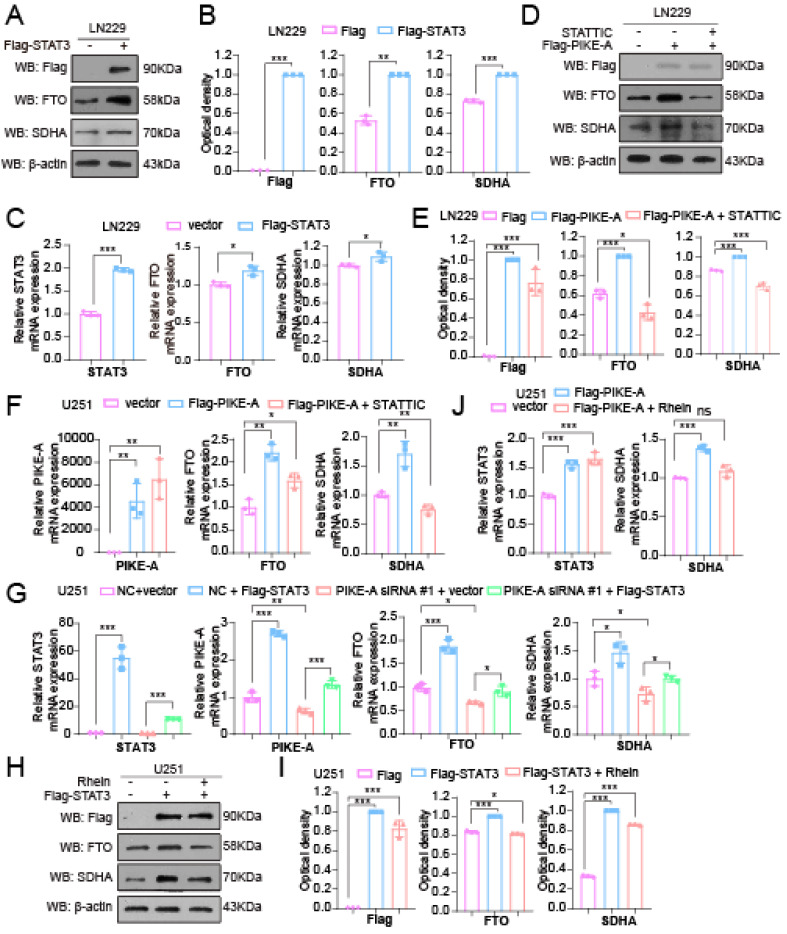
PIKE-A increases SDHA expression through activating FTO expression in a STAT3-dependent manner. (**A**,**B**) STAT3, FTO, and SDHA protein were determined by Western blotting in control and STAT3 overexpressed LN229 cells (**A**). The optical density of blotting bands was quantified by Image J software and normalized to actin (n = 3). The *p* values were calculated using Student’s *t* test (**B**). (**C**) STAT3, FTO, and SDHA mRNA levels were determined by qRT-PCR in control and STAT3 overexpressed LN229 cells (n = 3). The *p* values were calculated using Student’s *t* test. (**D**,**E**) PIKE-A, FTO, and SDHA protein levels were determined by Western blotting in PIKE-A overexpressed cells treated with STATTIC (**D**). The optical density of blotting bands was quantified by Image J software and normalized to actin (n = 3). The *p* values were calculated using Student’s *t* test (**E**). (**F**) PIKE-A, FTO, and SDHA mRNA levels were determined by qRT-PCR in PIKE-A overexpressed cells treated with STATTIC (n = 3). The *p* values were calculated using ANOVA. (**G**) PIKE-A, STAT3, FTO, and SDHA mRNA levels were determined by qRT-PCR in PIKE-A knockdown cells with or without exogenously expressing STAT3 (n = 3). The *p* values were calculated using ANOVA. (**H**,**I**) PIKE-A, FTO, and SDHA protein levels were determined by Western blotting in STAT3 overexpressed cells treated with Rhein (**H**). The optical density of blotting bands was quantified by Image J software and normalized to actin (n = 3). The *p* values were calculated using ANOVA (**I**), (**J**) STAT3 and SDHA mRNA levels were determined by qRT-PCR in STAT3 overexpressed cells treated with Rhein (n = 3). The *p* values were calculated using ANOVA. Error bars represent mean values ± SD from three independent experiments ( ns: not significant; *: *p* < 0.05; **: *p* < 0.01; ***: *p* < 0.001).

**Figure 5 ijms-23-11304-f005:**
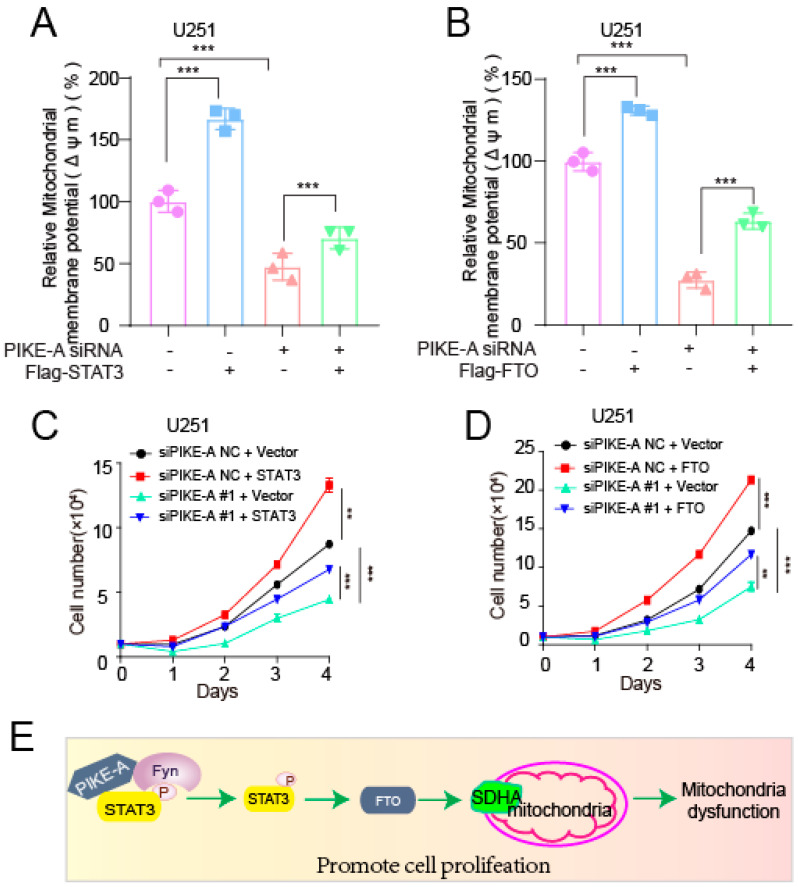
PIKE-A regulates mitochondrial respiration by regulating FTO expression in a STAT3-dependent manner. (**A**,**B**) Mitochondrial membrane potential was determined in PIKE-A knockdown cells with or without exogenously expressing STAT3 (**A**) or STAT3 (**B**) (n = 3). The *p* values were calculated using ANOVA. (**C**,**D**) Cell proliferation was determined by cell number counting assay in PIKE-A knockdown cells with or without exogenously expressing STAT3 (**C**) or FTO (**D**) (n = 3). The *p* values were calculated using ANOVA. (**E**) A schematic model shows that PIKE-A promotes glioblastoma growth by facilitating FTO/SDHA expression, which is dependent upon phosphorylated STAT3. Error bars represent mean values ± SD from three independent experiments (**: *p* < 0.01; ***: *p* < 0.001).

**Table 1 ijms-23-11304-t001:** The sequence of shRNA or siRNA.

Gene	shRNA Sequence
PIKE-A	CCGGGCTGTGATCAATAGCCAGGAACTCGAGTTCCTGGCTATTGATCACAGCTTTTTG
**Gene**	**siRNA Sequence**
PIKE-A	#1:GAGAAACGAAGCTTGGATACT
	#2:TGTGATCAATAGCCAGGAACT

**Table 2 ijms-23-11304-t002:** The antibodies.

Antibodies	Source	Identifier
FTO Rabbit mAb	proteintech	27226-1-AP
SDHA (D6J9M) XP^®^ Rabbit mAb	CST	11998
Bata Actin Mouse McAb	proteintech	66009-1-Ig
Flag Rabbit PolyAb	proteintech	20543-1-AP

**Table 3 ijms-23-11304-t003:** The sequence of real-time PCR primers.

Gene	Real-Time PCR Primers Sequence
PIKE-A	Forward primer: 5′- AGAGGCAGTTCGTTGTAGCT -3′
	Reverse primer: 5′- TCTGTCTTCTCCAGCACCTG -3′
STAT3	Forward primer: 5′- CAGCAGCTTGACACACGGTA -3′
	Reverse primer: 5′- AAACACCAAAGTGGCATGTGA -3′
FTO	Forward primer: 5′-AACACCAGGCTCTTTACGGTC-3′
	Reverse primer: 5′-TGTCCGTTGTAGGATGAACCC-3′
SDHA	Forward primer: 5′-CAAACAGGAACCCGAGGTTTT-3′
	Reverse primer: 5′-CAGCTTGGTAACACATGCTGTAT-3′

## Data Availability

All data will be made available upon reasonable request by emailing the corresponding author.

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
