# Peer review of "PIKE-A Modulates Mitochondrial Metabolism through Increasing SDHA Expression Mediated by STAT3/FTO Axis"

_ijms, 2022, doi:10.3390/ijms231911304_

Round 1

Reviewer 1 Report

In this manuscript Sun et al. investigate the role of PIKE-A in the energy metabolism in cancer. They find that PIKE-A increases the efficiency of the electron transport chain via SDHA expression. They further perform experiments to conclude that this mechanism is going via the STAT3/FTO axis. While the experimental design is clear and straightforward, the authors’ conclusions are not fully supported by the presented data and the manuscript includes several scientific sloppy parts  

The statistical analyses is not performed correctly and is missing important information to further interpret the data. The authors claim that they use student’s t test; but several graphs contain more than 2 variables (e.g. 3A, 3D, 3I,…) and thus for those other statistical tests should be applied. Furthermore, the authors claim that the groups are having similar variances. Did they test this? This is important to define if the parametric one way anova can be applied or the non-parametric Kruskal wallis. To test for normal distribution, a requirement for one way anova, a number of repetitions need to be performed (often starting at an n of 30); however, the authors did not include the number of repetitions for the performed experiments. For figures 4A-B, this is a grouped analyses and a two-way anova (or the nonparamtric alternative) should be applied as statistical analyses. In addition, it would be helpful to have the single data points represented in the graphs. In addition, quantification for western blotting analyses is lacking, so it is difficult to interpret the blots. Especially as for Fig3A siRNA 2 the siRNA does not seem to be active, while looking at the western blot in 3B, there does seem to be an effect on FTO and SDHA. This might suggest that the observed effects are not specific for PIKE-A knockdown.

A main concern is that the authors claim that PIKE-A increases SDHA expression through activating FTO expression in a STAT3-dependent manner, but the manuscript does not show any direct link. For instance, they do not show any data on the effect of PIKE-A expression or FTO expression on STAT3, even though they do mention STAT3 antibodies (also phosphor-stat3) in the methods section. Further in this line of thought (for figure 4), expression of STAT3 on its own already strongly increases mitochondrial membrane potential or cell number, so it seems logical that therefore there is also an improvement when PIKE-A is knocked down compared to only PIKE-A knockdown, there does not seem to be a direct link for that, the same but to a lesser extent holds true for FTO expression.

The usage of different cell lines is seemingly done in a random fashion. In the first figure all experiments are performed in two different cell lines (U251 and LN229), but in the latter figures this consistency is dropped and certain experiments are performed in one cell line while the follow-up experiments are performed in the other cell line. In addition, the methods also mention 293T cells, but it is unclear if they were used for something else than the creation of the stable knockdown cell line HEK293T.

The authors use the FTO inhibitor Rhein and show lowers FTO mRNA expression levels, but Rhein is known to competitively bind to the active site. Can the authors explain why they see a reduction in FTO mRNA expression? Also Rhein is also a deacetlyse, can the authors exclude possible off target affects to play a role?

The authors should make it clear which cells are used for fig 4A and 4B

The logical reasoning to go from mitochondrial respiration via SDHA regulation to methylation is not clear and should be further elaborated on in the initial part of 2.3

The western blots in 3E and 3H should be improved

The authors should clarify what the difference between PIKE-A siRNA 1 and 2 is, as in the methods section the siRNA sequence is the same

The authors need to carefully revise the manuscript for language and style. A few examples (limited to the introduction) but far from complete are below:

Line 36: space between binding and signal

Line 41: verb missing

Line 43: space between critical and role

Line 44: space between tumors and occurrence

Line 48: this are should be changed to these are or this is

Line 49: space following the end of the sentence

Line 52: verb missing and space following the end of the sentence

Line 54: space between enzyme and is and missing word

Line 55: space between been and demonstrated

Author Response

Review#1

In this manuscript Sun et al. investigate the role of PIKE-A in the energy metabolism in cancer. They find that PIKE-A increases the efficiency of the electron transport chain via SDHA expression. They further perform experiments to conclude that this mechanism is going via the STAT3/FTO axis. While the experimental design is clear and straightforward, the authors’ conclusions are not fully supported by the presented data and the manuscript includes several scientific sloppy parts. 

  1. The statistical analyses is not performed correctly and is missing important information to further interpret the data. The authors claim that they use student’s t test; but several graphs contain more than 2 variables (e.g. 3A, 3D, 3I,…) and thus for those other statistical tests should be applied. Furthermore, the authors claim that the groups are having similar variances. Did they test this?.This is important to define if the parametric one way anova can be applied or the non-parametric Kruskal wallis. To test for normal distribution, a requirement for one way anova, a number of repetitions need to be performed (often starting at an n of 30);however, the authors did not include the number of repetitions for the performed experiments. For figures 4A-B, this is a grouped analyses and a two-way anova (or the nonparamtric alternative) should be applied as statistical analyses.In addition, it would be helpful to have the single data points represented in the graphs.In addition, quantification for western blotting analyses is lacking, so it is difficult to interpret the blots.Especially as for Fig3A siRNA 2 the siRNA does not seem to be active, while looking at the western blot in 3B, there does seem to be an effect on FTO and SDHA. This might suggest that the observed effects are not specific for PIKE-A knockdown.

Response: We appreciate the reviewer for pointing out this important issue, we have revised the statistical analyses in the revised manuscript and indicated the statistical analyses and the number of repetitions in the figure legend (The differences between two groups with similar variances were analyzed using a two-tailed Student`s t test., For comparison among more than two groups were determined by one-way analysis of variance (ANOVA) with Tukey`s post hoc test). 

Moreover, we have reanalyzed the data as suggested that single data points represented in the graphs.

We performed quantification analyses for western blotting in the revised figures.

In addition, We have repeated experiments to improve the condition for the qRT-PCR assay to detect the knockdown efficiency of PIKE-A siRNA in Fig3A and found that the siRNA#2 also had knockdown effect. 

  1. A main concern is that the authors claim that PIKE-A increases SDHA expression through activating FTO expression in a STAT3-dependent manner, but the manuscript does not show any direct link. For instance, they do not show any data on the effect of PIKE-A expression or FTO expression on STAT3, even though they do mention STAT3 antibodies (also phosphor-stat3) in the methods section.

Response: We appreciate the reviewer for pointing out this important issue. We are sorry that we not make the logic clearly, we have add the schematic model in the revised figures. A schematic model shows that PIKE-A promotes glioblastoma growth by facilitating FTO/SDHA expression, which is dependent upon phosphorylated-STAT3.

In our previous study, we have confirmed that knockdown of PIKE-A reduces the level of p-STAT3(PMID: 34411567). Thus, in this study, we did not performed the assay again, and we have revised the manuscript to make the story clearly.

We are sorry about that we listed the STAT3 and phosphor-stat3 antibodies in the method parts, now we removed these information. 

  1. Further in this line of thought (for figure 4), expression of STAT3 on its own already strongly increases mitochondrial membrane potential or cell number, so it seems logical that therefore there is also an improvement when PIKE-A is knocked down compared to only PIKE-A knockdown, there does not seem to be a direct link for that, the same but to a lesser extent holds true for FTO expression.

Response: We appreciate the reviewer for pointing out this important issue. we are sorry that we did not describe it clearly in the manuscript. Mechanistically, we demonstrated that PIKE-A could regulate FTO expression via STAT3 and subsequently affect SDHA expression. Here, we wanted to demonstrate that PIKE-A could affect mitochondrial function via STAT3/FTO. As you said that STAT3 on its own already strongly increases mitochondrial membrane potential or cell number, that is ture. Activating STAT3 signaling promotes proliferation and inhibits apoptosis in cancer, thus, exogenous expression of STAT3 promotes cancer growth. We also got the same results that expression of STAT3 in control cells strongly increases mitochondrial membrane potential or cell number.

We are sorry that we not make the logic clearly, we have add the schematic model in the revised figures. A schematic model shows that PIKE-A promotes glioblastoma growth by facilitating FTO/SDHA expression, which is dependent upon phosphorylated-STAT3.

  1. The usage of different cell lines is seemingly done in a random fashion. In the first figure all experiments are performed in two different cell lines (U251 and LN229), but in the latter figures this consistency is dropped and certain experiments are performed in one cell line while the follow-up experiments are performed in the other cell line. In addition, the methods also mention 293T cells, but it is unclear if they were used for something else than the creation of the stable knockdown cell line HEK293T.

Response: We appreciate the reviewer for pointing out this important issue. We are very sorry that we did not use the same cell line in Figure 3 and Figure 4, because we consider that in molecular biology experiments, cell lines were merely used as instrumental cells, and in functional experiments, we used different cell lines.

In the present study, 293T cells were only used to packe shRNA virus for constructing PIKE-A stable knockdown U251 and LN229 cells, we haved revised method parts to make it clearly.

The short hairpin RNA (shRNA) plasmids targeting PIKE-A were purchased from Transheep Biological Corporation (Transheep, Shanghai, China). To establish stable knockdown cells, lentiviral shRNA constructs were co-transfected with viral packaging plasmids (psPAX2 and pMD.2G) into HEK293T cells. 24 hours after transfection, cell supernatants were collected on two consecutive days and then filtered through a 0.45 μm filter. U251 and LN229 cells were infected with the Lentivirus for 24 hours and selected with 2μg/mL puromycin. The knockdown efficacity was determined by qRT-PCR.

  1. The authors use the FTO inhibitor Rhein and show lowers FTO mRNA expression levels, but Rhein is known to competitively bind to the active site. Can the authors explain why they see a reduction in FTO mRNA expression?Also Rhein is also a deacetlyse, can the authors exclude possible off target affects to play a role?

Response: We appreciate the reviewer for pointing out this important issue. Our previous study (Clin Transl Med. 2022 Mar;12(3):e772) and Dr Yang`s studies (He develope Rehin as FTO inhibitor ) have shown that Rhein treatment decreased FTO protein expression. Here, we found that Rhein treatment decreased FTO mRNA levels, we explain that FTO regulates various target genes expression. While, several target genes also regulate FTO expression. Thus, we found that Rhein treatment decreased FTO mRNA levels, which may be due to the feedback by its target gene. In the future, we will explore this study.

  1. The authors should make it clear which cells are used for fig 4A and 4B

Response: We appreciate the reviewer for pointing out this important issue. We are sorry for that we did not make it clear in the manuscript and we have relabeled the figures.

  1. The logical reasoning to go from mitochondrial respiration via SDHA regulation to methylation is not clear and should be further elaborated on in the initial part of 2.3.

Response: We appreciate the reviewer for pointing out this important issue. We have revised the manuscript.

N6-methyladenosine (m6A) is the most abundant methylation modification on mammalian messenger RNA[23].Fat-mass and obesity-associated protein (FTO), the first RNA demethylase,is involved in the regulation of the major regulatory function in mammals via the regulation of m6A modifications[24]. For example, our previous studies show that FTO can regulate the expression of 6PGD and G6PD by demethylating m6A modifacion on their mRNA  by demethylation[24, 25]. Thus, to identify the mechanism of PIKE-A in regulating SDHA expression, we explored whether PIKE-A regulates SDHA expression mediated by FTO.

  1. The western blots in 3E and 3H should be improved

Response: We highly appreciate the detailed valuable comments of our manuscript, We have repeated experiments to improve the condition for the western blotting assay. Furthermore, we also performed quantification analyses for western blotting in the revised figures.

  1. The authors should clarify what the difference between PIKE-A siRNA 1 and 2 is, as in the methods section the siRNA sequenceis the same.

Response: We appreciate the reviewer for pointing out this important issue, we are sorry for that we did not make it clear in the manuscript, we have revised the mistakes of siRNA sequence.

10.The authors need to carefully revise the manuscript for language and style. A few examples (limited to the introduction) but far from complete are below:

Line 36: space between binding and signal

Line 41: verb missing

Line 43: space between critical and role

Line 44: space between tumors and occurrence

Line 48: this are should be changed to these are or this is

Line 49: space following the end of the sentence

Line 52: verb missing and space following the end of the sentence

Line 54: space between enzyme and is and missing word

Line 55: space between been and demonstrated

Response: We appreciate the reviewer for pointing out this important issue, we have proof the whole manuscript and revised the mistakes.

Reviewer 2 Report

This is an interesting manuscript that identifies a novel pathway regulating mitochondrial function in glioblastoma cells.  Although the findings are novel. The reviewer have several concerns:

Provide the basal expression of PIKE-A in two cell lines and their mitochondrial function at the basal state. This will be important to correlate the PIKE-A regulates oxidative phosphorylation theory.

Please provide the absolute values for the mitochondrial functional measurements in figure 1.

PIKE-A knockdown was in LN229 cells and upregulation was done in U-251 why? Please provide the results of PIKE-A knockdown in U251 and upregulation in LN229.

Multiple typographical errors Lines 36, 44, 54, 55

SDH is used for both succinic acid dehydrogenase and succinate dehydrogenase please use only one.

Author Response

Review#2

This is an interesting manuscript that identifies a novel pathway regulating mitochondrial function in glioblastoma cells. Although the findings are novel. The reviewer have several concerns:

  1. Provide the basal expression of PIKE-A in two cell lines and their mitochondrial function at the basal state. This will be important to correlate the PIKE-A regulates oxidative phosphorylation theory.

Response: We appreciate the reviewer for pointing out this important issue,We are sorry for that we did not make it clear in figure, we have detected PIKE-A expression by qRT-PCR in U251 and LN229 cells, and we found that PIKE-A expression was higher in U251 cells than in LN229 cells.

  1. Please provide the absolute values for the mitochondrial functional measurements in figure 1.

Response: We appreciate the reviewer for pointing out this important issue,we are sorry for that we did not make it clear in figure, we have reanalyzed the original data of Mitochondrial membrane potential.

3.PIKE-A knockdown was in LN229 cells and upregulation was done in U-251 why? Please provide the results of PIKE-A knockdown in U251 and upregulation in LN229.

Response: We highly appreciate the detailed valuable comments of our manuscript. We performed PIKE-A knockdown in LN229 cells (Figure 1a, 1b, 2a,2b, 2e). 

We performed PIKE-A knockdown in U251 cells (Figure 1a, 1b, 2b, 3b,3c, 3d, 4g, 5a-5d ),

We performed exnogenous expression PIKE-A in LN229 cells (Figure 4d,4e,4f)

We performed exnogenous expression PIKE-A in U251 cells (Figure 3e,3f, 3g,3h,3i).

4.Multiple typographical errors Lines 36, 44, 54, 55

Response: We appreciate the reviewer for pointing out this important issue, we have proof the whole manuscript and revised these mistakes.

5.SDH is used for both succinic acid dehydrogenase and succinate dehydrogenase please use only one.

Response: We appreciate the reviewer for pointing out this important issue, we have changed succinic acid dehydrogenase to succinate dehydrogenase.

Reviewer 3 Report

In this manuscript, authors reported the role of PIKE-A, an activator of Akt pathway, in regulation of mitochondrial oxphos complex II component expression and its activity. Authors reported that PIKE-A is crucial for expression of SDHA in tested cell lines. There are several concerns with this manuscript as mentioned below-

1. There is no data presented in this manuscript which show that PIKE-A regulate Stat-3 pathway. Authors did overexpress stat-3 in cells and analyzed the expression of FTO and SDHA but without analyzing Stat-3 activation (p-Stat-3) and expression (Total Stat-3) in cells following PIKE-A knockdown, the manuscript title cannot be justified and misleading.

2. Given the fact that PIKE-A is a regulator of Akt pathway which is known to activate mitochondrial oxphos activity, its surprising that authors did not explore Akt pathway at all in this manuscript.

3. In Figure 3A, PIKE-A mRNA expression data clearly shows that there is no knockdown in expression of PIKE-A in siRNA#2 group still FTO and SDHA expression is downregulated as shown in Figure 3B. This is a serious discrepancy and should be addressed.

4. What happened to PIKE-A expression following Stat-3 overexpression and vice versa?

5. Another key experiment to justify the overall conclusion and title of the manuscript is to analyze the expression of FTO and SDHA after overexpressing Stat-3 in PIKE-A knockdown cells (similar groups shown in Figure 4A).

6. In Figure 2, there is no Panel F as mentioned in Figure legend.

7. Since authors emphasized too much on mitochondrial metabolism, ATP assay and Oxygen consumption should be analyzed along with mitochondrial membrane potential as shown in Figure 4A and 4B.

8. Table 1. Authors used only one PIKE-A shRNA construct in the paper but mentioned three shRNA sequences. Also, the sequences for PIKE-A siRNA#1 and #2 are same.

9. Table 2. There is no data of p-Stat-3, Stat-3 and ki-67 in the manuscript still these antibodies are mentioned in table 2.

10. Section 4.7. Authors mentioned that the analyzed the data using microscope then authors need to show some images and explain in material and methods how they quantify and presented the data in manuscript so that other researchers can follow the same for their own research as Flow cytometry is not available at many places.

Author Response

Review 3

In this manuscript, authors reported the role of PIKE-A, an activator of Akt pathway, in regulation of mitochondrial oxphos complex II component expression and its activity. Authors reported that PIKE-A is crucial for expression of SDHA in tested cell lines. There are several concerns with this manuscript as mentioned below-

  1. There is no data presented in this manuscript which show that PIKE-A regulate Stat-3 pathway. Authors did overexpress stat-3 in cells and analyzed the expression of FTO and SDHA but without analyzing Stat-3 activation (p-Stat-3) and expression (Total Stat-3) in cells following PIKE-A knockdown, the manuscript title cannot be justified and misleading.

Response: We appreciate the reviewer for pointing out this important issue, We have detected the p-STAT3 level in the previous study, found that the p-STAT3 was decreased in PIKE-A knockdown cells(PMID: 34411567).

  1. Given the fact that PIKE-A is a regulator of Akt pathway which is known to activate mitochondrial oxphos activity, its surprising that authors did not explore Akt pathway at all in this manuscript.

Response: We appreciate the reviewer for pointing out this important issue, In Dr Ye's studies have shown that compared to control cells, robust Akt phosphorylation is observed in wild-type PIKE-A infected cells (PMID: 15118108). In this study, we mainly confirmed that PIKE-A regulates SDHA expression through STAT3/FTO, so we did not detect p-Akt.

  1. In Figure 3A, PIKE-A mRNA expression data clearly shows that there is no knockdown in expression of PIKE-A in siRNA#2 group still FTO and SDHA expression is downregulated as shown in Figure 3B. This is a serious discrepancy and should be addressed.

Response: We appreciate the reviewer for pointing out this important issue. We have repeated experiments to improve the condition for the qRT-PCR assay to detect the knockdown efficiency of PIKE-A siRNA in Fig3A and found that the siRNA#2 also had knockdown effect. 

  1. What happened to PIKE-A expression following Stat-3 overexpression and vice versa?

Response: We appreciate the reviewer for pointing out this important issue. We found that exnogenous expression of STAT3 increased the PIKE-A mRNA levels in PIKE-A knockdown control cells (Figure 4F). Again, we also found that exnogenous expression of PIKE-A increased the STAT3 mRNA levels in PIKE-A knockdown control cells (Figure 4H).

  1. Another key experiment to justify the overall conclusion and title of the manuscript is to analyze the expression of FTO and SDHA after overexpressing Stat-3 in PIKE-A knockdown cells (similar groups shown in Figure 4A).

Response: We appreciate the reviewer for pointing out this important issue. we are sorry that we did not describe it clearly in the manuscript. Mechanistically, we demonstrated that PIKE-A could regulate FTO expression via STAT3 and subsequently affect SDHA expression. Here, we wanted to demonstrate that PIKE-A could affect mitochondrial function via STAT3/FTO.

In order to make the logic clearly, we have add the schematic model in the revised figures. A schematic model shows that PIKE-A promotes glioblastoma growth by facilitating FTO/SDHA expression, which is dependent upon phosphorylated-STAT3.

Thus, to analyze the expression of FTO and SDHA after overexpressing Stat-3 in PIKE-A knockdown cells is reasonable.

  1. In Figure 2, there is no Panel F as mentioned in Figure legend.

Response: We highly appreciate the detailed valuable comments of our manuscript, we are sorry for that we did not make it clear and we have deleted it.

  1. Since authors emphasized too much on mitochondrial metabolism, ATP assay and Oxygen consumption should be analyzed along with mitochondrial membrane potential as shown in Figure 4A and 4B.

Response: We highly appreciate the detailed valuable comments of our manuscript, Our school has no relevant detecting conditions, and we plan to go to other schools to detect the OCR and ATP. However, due to the Covid-19, we were locked in the school, and we are very sorry for not being able to supplement the experiment. We believe that the indicator of mitochondrial membrane potential is also representative of mitochondrial respiration; therefore, we dropped the data on OCR and ATP in Figures 1B and 1D simultaneously.

  1. Table 1. Authors used only one PIKE-A shRNA construct in the paper but mentioned three shRNA sequences. Also, the sequences for PIKE-A siRNA#1 and #2 are same.

Response: We appreciate the reviewer for pointing out this important issue, we are sorry for that we did not make it clear in manuscript, and we have revised the mistakes of shRNA and siRNA sequence.

  1. Table 2. There is no data of p-Stat-3, Stat-3 and ki-67 in the manuscript still these antibodies are mentioned in table 2.

Response: We appreciate the reviewer for pointing out this important issue, we have proof the whole manuscript and revised the mistakes.

  1. Section 4.7. Authors mentioned that the analyzed the data using microscope then authors need to show some images and explain in material and methods how they quantify and presented the data in manuscript so that other researchers can follow the same for their own research as Flow cytometry is not available at many places.

Response: We appreciate the reviewer for pointing out this important issue, we are sorry for that we did not make it clear in manuscript. JC-1 is an ideal fluorescent probe widely used to detect the mitochondrial membrane potential ∆ψm. At high mitochondrial membrane potential, JC-1 accumulates in the matrix of mitochondria and forms a polymer (J-aggregates), which can produce red fluorescence. When the mitochondrial membrane potential is low, JC-1 cannot accumulate in the matrix of mitochondria. At this time, JC-1 is monomer and can produce green fluorescence. This makes it very convenient to detect changes in mitochondrial membrane potential by the fluorescence color transition. Therefore, we used a fluorescence microscope to take photos, quantified and statistically analyzed the data using Image J software.

Round 2

Reviewer 1 Report

The authors have done a great job of improving the manuscript. One major issue concerning the statistical analyses remains, though. The sample size of n = 3 is too small to perform parametric statistical analyses such as t test or ANOVA; as the assumption of normally distributed data cannot be fulfilled in this small sample size.  Non-parametric statistical analyses will not reach significance with a sample size of n=3. In addition, it is questionable that a generalization can be robust that is based on such few data points as a statistical test aims to generalize from observed data to a broad underlying general population

Author Response

Response: We appreciate the reviewer for pointing out this important issue. we are sorry that we did not make it clearly in the manuscript. We usually think that three independent repeated experiments in molecular biology experiments are sufficient to prove relevant problems. Moreover, we used different cell lines for verification in the experiments, and corresponding conclusions can also be drawn. Therefore, our experimental data are independent three repeated experiments. In addition, we used GraphPad Prism software for data analysis and statistics to obtain the corresponding statistical results. In addition, we quoting an article published in 《Cell》 and 《advanced science》, is the same. As follows:

(I and J) qPCR analysis of transcripts involved in glutathione metabolism. (I) or diverse transcripts that have been identified as BCL2 effector protein dependent. (J) of PT and PS generated from PC9 CYCS−/− cells, reconstituted ± WT CYCS. n= 3 samples are shown. Statistical analysis was performed using unpaired Student’s t test. ∗∗p < 0.01, ∗∗∗p < 0.001, and ∗∗∗∗p < 0.0001. Sublethal cytochrome c release generates drug-tolerant persister cells.(PMID:36055199)

E,F) Rescue assays were performed in MC‐HBV‐transfected and LINC01431 overexpressed Huh7 cells after silencing PRMT1 or in the presence of the PRMT1 inhibitor C‐7280948 (12.8 µm). The enrichment of PRMT1, H4R3me2a, and epigenetic modifications on cccDNA were measured using indicated antibodies. For all experiments, representative of 3 independent experiments. Data information: data were presented as mean ± SD and normalized to the control group. One‐way ANOVA (C,E,F). LINC01431 Promotes Histone H4R3 Methylation to Impede HBV Covalently Closed Circular DNA Transcription by Stabilizing PRMT1.(PMID:35398991)

Reviewer 3 Report

Authors responded to all the raised concerns satisfactorily and manuscript may be accepted for publication.

Author Response

Thanks

Round 3

Reviewer 1 Report

I appreciate the authors' comments, but from a statistical point of view they are not correct.